# Connective tissue inspired elastomer-based hydrogel for artificial skin via radiation-indued penetrating polymerization

Yuan Tian[1], Zhihao Wang[1], Shuiyan Cao[2], Dong Liu[3], Yukun Zhang[1], Chong Chen[1], Zhiwen Jiang [1,4], Jun Ma [1,4] ✉ & Yunlong Wang [1] ✉

Robust hydrogels offer a candidate for artificial skin of bionic robots, yet few hydrogels have a comprehensive performance comparable to real human skin. Here, we present a general method to convert traditional elastomers into tough hydrogels via a unique radiation-induced penetrating polymerization method. The hydrogel is composed of the original hydrophobic crosslinking network from elastomers and grafted hydrophilic chains, which act as elastic collagen fibers and water-rich substances. Therefore, it successfully combines the advantages of both elastomers and hydrogels and provides similar Young's modulus and friction coefficients to human skin, as well as better compression and puncture load capacities than double network and polyampholyte hydrogels. Additionally, responsive abilities can be introduced during the preparation process, granting the hybrid hydrogels shape adaptability. With these unique properties, the hybrid hydrogel can be a candidate for artificial skin, fluid flow controller, wound dressing layer and many other bionic application scenarios.

The rapid development of hydrogel has opened up a plethora of possibilities in the ever-growing field of bionic devices, including soft robots[1–3], artificial skin[4–6], wearable electronic devices[7–9], and biomedical engineering materials[10,11]. However, achieving high-performance hydrogels comparable to realistic human skin faces critical challenges that need to be urgently addressed. Human skin is soft (Young's modulus: 0.1–2 MPa), stretchable (140–180%), water-rich (cuticle moisture content of about 25%, the other parts close to 70%), and breathable biological organ[12]. Unfortunately, all available soft materials, including elastomers and hydrogels, can only partially meet these requirements. While traditional elastomers display excellent mechanical properties, and environmental stability, hydrogels possess unique properties such as high water content, good biocompatibility, breathability, and responsiveness to chemical environment[13–15].

In the last decade, many efforts have been put into imitating the ingenious design of connective tissues found in nature. As soft, robust, highly adjustable, and water-rich hydrogels, connective tissues are used to compose various parts of the human body ranging from ultra-soft tissues such as subcutaneous tissue to stiffer tissues such as skin and even robust cartilage. The functionality is attributed to the soft biological structure consisting of elastic collagen fibers, water-rich ground substances, and cells that generate the two components above. The elastic collagen fibers give them high elasticity and tough mechanical properties, while the ground substance provides hydrophilic properties. Despite considerable progress, critical challenges still exist to find an effective and scalable way to create hydrogels with comprehensively good performance, including elasticity, toughness, resistance to stress, biocompatibility, and a similar friction coefficient to that of real skins.

[1]College of Materials Science and Technology, Nanjing University of Aeronautics and Astronautics, Nanjing 211106, Jiangsu, China. [2]College of Physics, MIIT Key Laboratory of Aerospace Information Materials and Physics, Nanjing University of Aeronautics and Astronautics, Nanjing 211106, Jiangsu, China. [3]Key Laboratory of Neutron Physics and Institute of Nuclear Physics and Chemistry, China Academy of Engineering Physics, Mianyang 621999, Sichuan, China. [4]School of Nuclear Science and Technology, University of Science and Technology of China, Hefei 230026, Anhui, China. ✉e-mail: junma@nuaa.edu.cn; wylong@nuaa.edu.cn

Mimicking connective tissues can be achieved through various strategies, the most common and efficient of which involve the use of double network (DN) hydrogels and phase separation of distinct polymers[16–18]. DN hydrogel uses one highly crosslinked polyelectrolyte network to provide a rigid bracket, and the other specific lowly crosslinked or non-crosslinked neutral chain to absorb external stress[19]. Phase separation occurs when the hydrophobic polymeric chain or domains are introduced to form aggregates that act as physical crosslinking points, while also inhibiting the hydration of polymer chains, thus providing enlarged cohesive energy[20–22]. Polyampholytes are used to inhibit static electricity in hydrogels, which can also increase hydrophobicity and enhance crosslinking networks[23–25]. From the existing literature, it is evident that hydrophobic chain/domain formed heterogeneous structures are critical for the toughness and fatigue resistance of hydrogels; however, there are many limitations to introduce them in the practical preparation of hydrogels, the most important of which is the solubility of precursors[26,27]. Unfortunately, the practical fabrication of such heterogeneous hydrogels has been greatly limited by the difficulty of forming pre-solution containing monomers or precursors of elastomers before initiating the polymerization process, as these monomers or precursors are often hydrophobic and insoluble in water[26]. As a result, the commonly used elastomers or their monomers, such as isoprene, butyl acrylate natural rubber, or polydimethylsiloxane are often excluded from the preparation of hydrogels.

Predictably, the elastomers and hydrogel can serve as elastic collagen fibers and water-rich ground substance, forming connective tissues-like hydrogels that break down the barriers between these two very different materials, thus creating better candidate materials for artificial skin[15,26,28–30]. However, most literature has focused on the surface modification or macroscopic combining of elastomers and hydrogels through enhanced adhesion, involving multi-step complicated chemical modification and coupling reactions[15,27,31,32]. Yet, a homogenous composite of a water-swellable hydrogel with a hydrophobic backbone such hydrogels has never been reported. Therefore, there is an urgent need to develop new materials to meet the increasing demand for high-performance soft materials in bionic applications. Fortunately, γ-ray induced polymerization of hydrogels, which has been widely used in hydrogel wound dressing[33], can provide a modification method because it has an unmatchable penetrating ability to generate radicals in the interior region of hydrophobic elastomers and thus make uniform distribution of grafted chains across the elastomers possible[34]. Besides, this highly versatile process has few limitations in selecting base polymers or monomers, and the adjustable parameters provide the freedom to customize the material's composition and structure. Therefore, this design that combines the advantage of elastomers and hydrogels can be realized by adjusting the reaction system and radiation processing conditions according to the well-known grafting front mechanism in the radiation-induced modification of membranes[34].

Here, we report a straightforward and versatile approach to directly convert commercial elastomer i.e., silicone rubber into connective-tissue-inspired elastomer-based hybrids (CEBH) through a one-step radiation grafting technique. Benefiting from the power of radiation-induced modification, this synthesis process does not require any biologically toxic reagents or solvents, nor any crosslinking agents, coupling agents or photoinitiators, thus ensuring the biocompatibility of the obtained CEBH, which is essential for their applications as biomedical materials or artificial skins. The CEBH exhibits excellent mechanical properties, ion sensitivity and similar adhesion to human skin. We further demonstrated their applications in wound dressings, controllable flow conduits, shape memory-based fracture connection, and cladding materials of manipulators. Moreover, this method is generally applicable to various elastomers, including polydimethylsiloxane, natural latex, and VHB. We anticipate that this newly developed preparation method of CEBH can provide a simple, scalable, and general method to prepare newly structured hydrogels that combines the elastomers and hydrogels, and offers potential artificial-skin materials with comprehensive mechanical and surface properties comparable to real human skin, thus promoting their applications in soft robots, artificial skin, wearable electronic devices, and biomedical engineering.

## Results

### The design and micro-structure of CEBH

In contrast to the existing methods, the CEBH preparation we proposed starts from commercially available, fully crosslinked elastomers, such as highly crosslinked silicone rubber, which possess distinct properties and behave differently from hydrogels in all aspects. Through radiation-indued penetrating polymerization processing, the elastomers become highly swollen in water and exhibit excellent biocompatibility. The CEBH successfully bridges the gap between hydrogels and elastomers, combining the advantages of each. The basic concept of this method is illustrated in Fig. 1a. In order to imitate structures that compose the connective tissues, a two-component polymer chain network was designed, where the PDMS forms the elastic crosslink network, and the PAA grafted on the backbone serves as a hydrophilic substrate to store and exchange water. In the natural connective tissues, these functionalities are attributed to the elastic collagen fibers and water-rich ground substances, respectively. To accomplish this design, the well-crosslinked elastomers were chosen as the beginning of the preparation. The elastomer (e. g. silicon rubber) was immersed in an aqueous solution of acrylic acid (AAc) containing a small amount of Mohr's salt, and then the reaction vessel was purged with an inert gas to expel oxygen before being irradiated with γ-ray for a certain period in a $^{60}$Co source. As the absorbed doses used for the modification process increased, it can be observed that the silicone rubber is grafted with polyacrylic acid (PAA), and the modified silicone rubber first becomes hydrophilic and then fully swollen in water after the degree of grafting surpasses a critical value. With the increment of polyacrylic acid content (AC), the hydrophilicity of the CEBH gradually increased, which is similar to the traditional radiation grafting modifications, and the contact angle decreases accordingly (Fig. 1b). Interestingly, as the grafting continues, the formed composite can swell in water and acts as a typical hydrogel. As shown in Supplementary Fig. 1a, b, when the PAA content in the grafts increased to 72%, the modified silicone rubber can swell more than 3 times its original size in water, indicating that the water content in the swollen material is over 90%, demonstrating that the silicone rubber has been successfully converted into a hydrogel from the inside out. Furthermore, the radiation-indued penetrating polymerization processing is compatible with various commercial elastomers. Natural latex, fluorine gum, polyurethane, styrene-ethylene-butylene-styrene (SEBS) and VHB have also been treated through a similar process, and the results indicate that we provide a straightforward and universal method to prepare CEBH from various commercial elastomers, as shown the Supplementary Fig. 1b–d.

The effect of monomer concentration, absorbed doses, as well as Mohr's salt (radical scavenger, added to inhibit the homopolymerization process in the solution) concentration on the AC was investigated, as shown in Supplementary Fig. 2a–c. It can be concluded that with the increase of the monomer concentration and absorbed doses, the AC grows accordingly and then reaches a plateau, while only a moderate concentration of ferric ions can achieve the maximum AC. As reported in previous literature[35], these trends can be attributed to the gel effect and inhibition of homopolymerization of AAc monomer by the ferric ions. In fact, it is impossible to perform even grafting of AAc without Mohr's salt because of the strong homopolymerization of AAc, not to mention the penetrating grafting polymerization. The selective inhibition of homopolymerization is recognized by the difficulty of

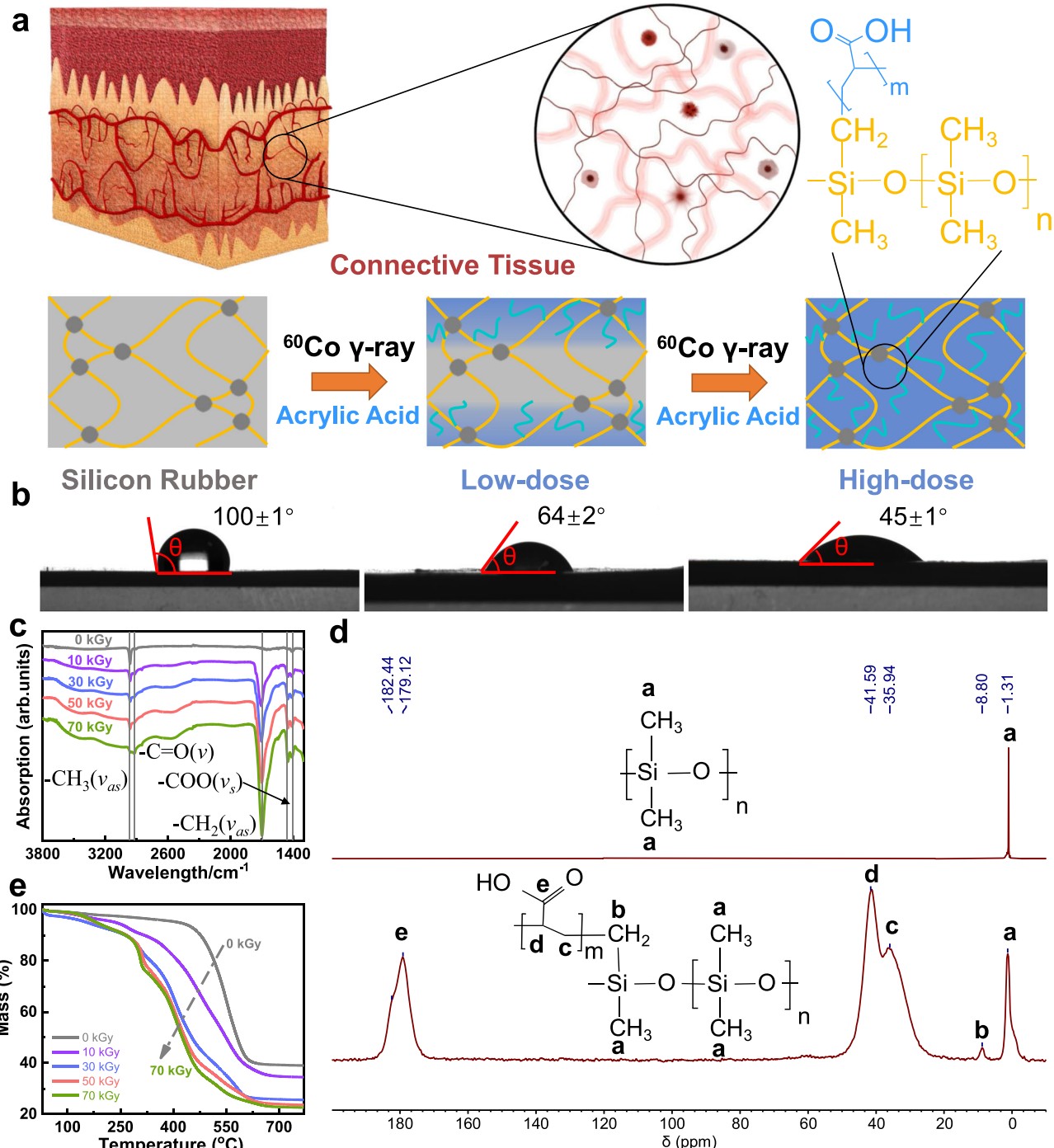

**Fig. 1 | Design and structure characterization of CEBH. a** The concept diagram of silicone rubber modified by acrylic acid into CEBH. **b** Experimental optical diagram of contact angle of modified silicone rubber with different AC. **c** FT-IR spectra of CEBH prepared at different absorbed doses. **d** The solid-state $^{13}$C-NMR of silicone rubber (upper) and CEBH (lower). **e** Thermogravimetric spectrum of CEBH, the absorbed doses applied in the preparing process are 0, 10, 30, 50, and 70 kGy.

$Fe^{2+}/Fe^{3+}$ diffusion to the surface of the polymer base, and the complex of the $Fe^{2+}/Fe^{3+}$ with PAA[36,37]. Furthermore, this process has many advantages over chemical grafting methods, such as being free of initiators, being conducted at room temperature and having a simple operation. However, compared with surface modification, this grafting process is derived from the penetration of monomers into the polymer matrix, making the grafting process a bulk grafting instead of obtaining surface grafts on the silicone rubber, which has been reported by many other works[38,39]. Additionally, the degree of grafting, as well as the swelling properties of prepared CEBH can be easily adjusted by the conditions used in the modification process.

Fourier transform infrared spectroscopy (FT-IR), thermal gravity (TG), $^{13}$C-NMR spectroscopy, and scanning electron microscope (SEM) were used to give a comprehensive understanding of the prepared CEBH. As depicted in Fig. 1c, FT-IR spectra of CEBH with different AC were shown, verifying the increment of PAA content. The spectra showed the presence of characteristic peaks of −COOH at 1702 cm$^{-1}$, and 2962 cm$^{-1}$ as the antisymmetric stretching peak of −CH bonds, indicating that silicone rubber is successfully grafted with AAc after the radiation processing. As shown in Fig. 1d, the $^{13}$C-NMR spectroscopy of the dried hydrogel also proved the success of grafting. The peak at a

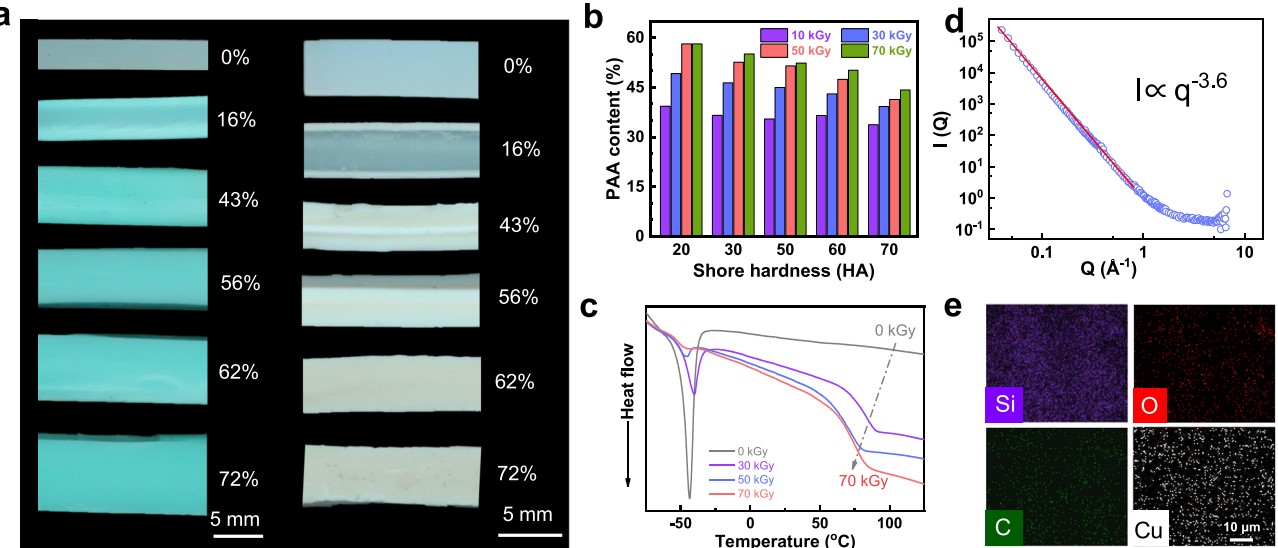

**Fig. 2 | Penetration-induced polymerization and phase structure of CEBH.** **a** Digital images of CEBH with different AC after being swollen in 1 g/mL Cu²⁺ solution (left) and toluene (right). **b** The effect of silicone rubber hardness on the AC of CEBH. **c** The DSC spectra of CEBH. **d** Neutron scattering spectra of CEBH AC of 62% fully swollen in deuterated water. **e** The EDS mapping of CEBH with an AC of 30%.

chemical shift of 8.80 clearly shows the presence of the carbon atom of AAc grafted -O-Si(CH₂-g-PAA)- was shifted to a higher field, proving the grafting of AAc as a branch onto the PDMS chain network via covalent bonds.

As shown in Fig. 1e and Supplementary Fig. 2d, the original silicone rubber began to sublimate at 500 °C, while the CEBH has an additional weight loss process beginning at approximately 300 °C, and the weight loss remains proportional to the graft ratio. The first step (before 200 °C) is attributed to the loss of water; the second step (235–329 °C), is attributed to the formation of PAA anhydride; the third step (331–512 °C) is attributed to the degradation of the corresponding PAA anhydride; the final step (up to 693 °C) is attributed to the thermal degradation of backbone. As the content of PAA increases, the bound water will increase, and the self-association of carboxyl groups will increase, which is beneficial to the second and third weight loss steps. Considering that the decarboxylation and main chain fracture of PAA were reported in a similar temperature region, this weight loss process accounts for the weight loss of PAA grafted in the composite[40]. Moreover, SEM images of the cross-section showed no obvious microphases, implying that the grafting of AAc was homogeneous in the silicone rubber matrix without phase separation, as depicted in Supplementary Fig. 2e.

The adsorption capacity of CEBH with different AC for toluene and water was illustrated in Supplementary Fig. 3a, b. The results indicate that unmodified silicone rubber exhibits a large adsorption capacity for toluene and is highly hydrophobic. In contrast, CEBH has strong hydrophilicity and can swell in water. As shown in Fig. 2a and Supplementary Fig. 3c, the grafting front, with a clear layering phenomenon, passed through the entire thickness of silicone rubber when the PAA content exceeded 43%. Before that, the surface layer can swell in water, while the inner layer remains hydrophobic and water-resistant. As the degree of grafting increases, the thickness of hydrophilic layers keeps growing, while the hydrophobic core layer shrinks and eventually disappears. When immersed in toluene, unmodified silicon rubber expanded to absorb toluene, and the swelling ratio decreased with the increment of AC. These observations provided compelling evidence that the grafting process is controlled by the simultaneous penetration process of the AAc monomer.

The penetration during the grafting process can be described by a dimensionless factor, which can be mathematically expressed as

$$\alpha = \left[ \frac{k_P G_R \dot{D}^{0.5}}{k_i^{0.5} d} \right] \frac{L}{2}$$

where $k_i$ and $k_p$ are the kinetic rate constant of the initiation and polymerization process, $G_R$ is the $G$ value radicals generated by the radiation, $D$ is the absorbed dose rate applied in the grafting process, $d$ is the penetration parameter of the monomer into the polymer matrix, and $L$ is the thickness of the polymer matrix. Since $k_i$, $k_p$, and $d$ are constants, reducing the absorbed dose rate is the only way to enhance the penetration of the monomer. Typically, radiation grafting often initiates at the surface of the film and gradually progresses inward as the grafting zone swells from the monomer solution. Consequently, non-uniform graft copolymerization is frequently observed in the cross-section of the membrane with low grafting yield, following the well-known grafting front mechanism in the modification of membranes[34]. In elastomers, the grafting front can penetrate deeper as the crosslinking network is more easily expandable. Therefore, the macroscopic-scale elastomers can be grafted from the outside to the inside if the processing process is well controlled. In order to clarify the effect of entropic elasticity theory on the penetration of monomer, we conducted experiments using silicon rubbers with different hardness (also different crosslinking ratios) as model materials (Supplementary Fig. 3d). As shown in Fig. 2b and Supplementary Fig. 3e, increasing the crosslinking degree of silicone rubber resulted in a gradual decrease in the AC of CEBH. This suggests that the increasing contractile force provided by the polymer network suppresses the penetrating of monomer during the grafting process, thereby constraining the subsequent grafting process. As the absorbed radiation dose and dose rate were kept the same, the difference can be ascribed to the increasing difficulty of hydrophilic monomer penetration, which is caused by an elastic network with a higher crosslinking degree. As previous literature reported, the increment of contractile force will increase as the swelling ratio increases in elastomer or gels[41]. Therefore, the contractile force also increases when the hydrogels' swelling ratio increases with the AC, making further grafting in the out-layer hydrogel more difficult. As a result, the excessive grafting or swelling of the our-layer hydrogel was also

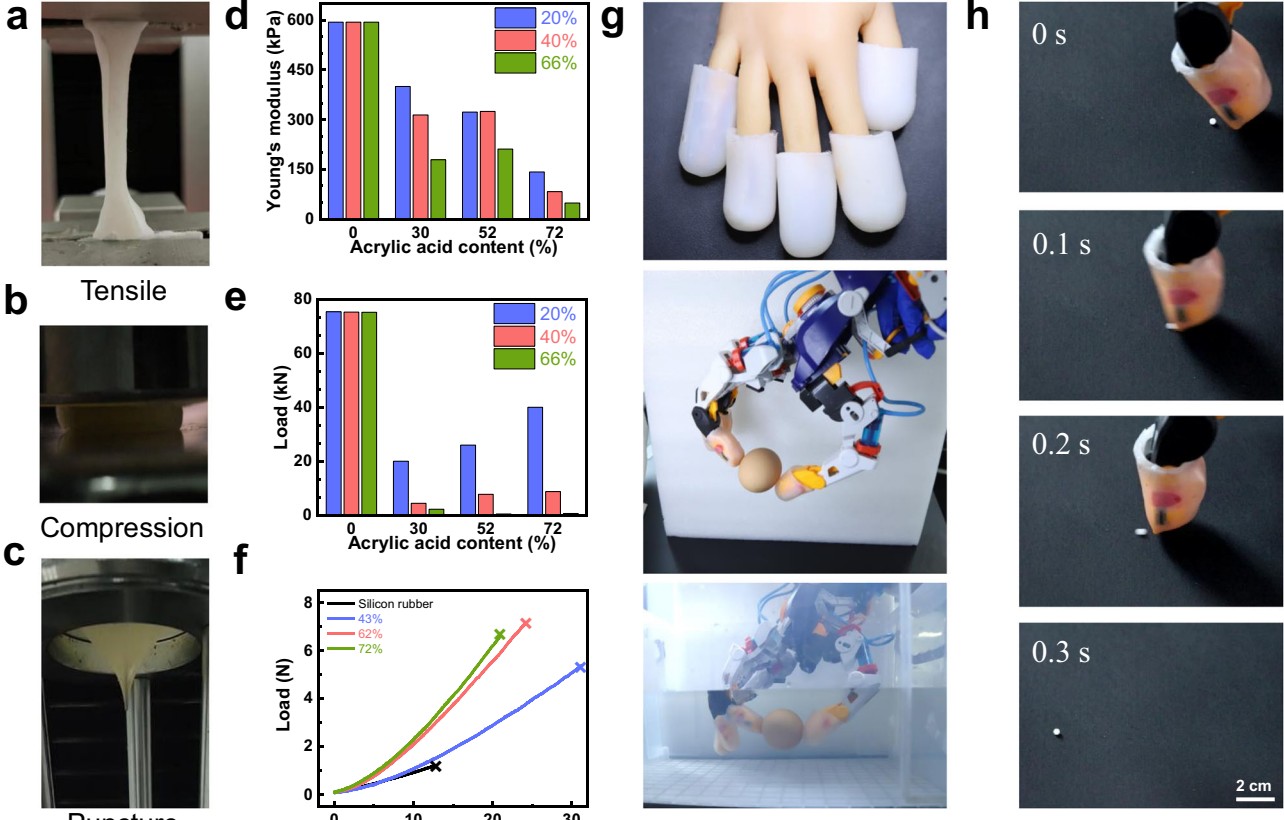

**Fig. 3 | The mechanical properties of CEBH and application as robot cladding materials. a–c** The mechanical properties characterization of CEBH, tensile tests, compression tests, and puncture tests. **d, e** The effect of AC on the Young's modulus and maximum pressure resistance of CEBH, the water content of CEBH is 20%, 40%, and 66%. **f** Force-displacement curve obtained by the puncture experiment of original silicon rubber and CEBH (with an AC of 43%, 62%, and 72%, the water content of CEBH is 20%). **g** The optical image of the egg-grabbing test using a CEBH covered robot arm, both in the air and underwater. **h** Flicking polystyrene foam beads (3 mm) using a CEBH covered robot arm. In (**g**), (**h**), the AC of CEBH on the surface of the robotic arm is 62% and is in a wet state.

constrained by the increasing contractile force, avoiding the inhomogeneity of the degree of grafting at different depths of the polymer matrix. The compelling evidence of the contractile force's impact was provided by the observed correlation between enhanced crosslinking and the inhibition of AC increment.

The CEBH grafted with AAc from the inside out can hardly swell in toluene, indicating that there is not a continuous hydrophobic phase in the CEBH. To further investigate the phase structure of the CEBH, characterizations including DSC, SEM, EDS mapping, and SANS were conducted. As shown in Fig. 2c, the glass transition temperature ($T_g$) of pure silicone rubber is −44 °C, while the $T_g$ of pure PAA is 106 °C, and the $T_g$ of CEBH is about 80 °C. With an increase of AC, the glass transition process at −44 °C (attributed to the silicon rubber) significantly declines, and the $T_g$ of CEBH slightly decreases. Moreover, the $T_g$ of CEBH was much smaller than that of pure PAA, indicating neither the silicone rubber nor PAA forms a continuous phase in the CEBH. The SANS scattering results in Fig. 2d showed no Guninier region in the detection region of the SANS spectrometer (Suanni), and the Porod fitting result gave a power-law slope of −3.6, indicating that the molecular chain in the swollen hydrogel showed a surface fractal structure with a fractal dimension of 2.4. Hence, the interface of deuterated water-swollen PAA and hydrophobic PDMS is highly interspersed at the molecular chain scale[42]. The missing Guninier region also suggests that there is no clear phase separation in the detection region[43], which can be as small as ~100 nm on the spectrometer Suanni[44]. Furthermore, we try to immerse the CEBH into Cu²⁺ solution and then characterize the elements' distribution via EDS mapping. However, as shown in Fig. 2e, the Cu, Si, C, and O elements

distribution in the CEBH indicates a homogeneous structure without domain structures. Therefore, the grafting of PAA on the silicone rubber matrix occurs at the molecule chain scale, forming an inter-penetrating network at the molecular chain scale, and the possibility of a bi-continuous phase structure can be completely excluded.

## Mechanical properties and friction coefficient of CEBH

In order to quantify the mechanical properties of the composite hydrogel prepared by this method, various mechanical tests were performed (Fig. 3a–c and Supplementary Movies 1, 2). We first used the standard tensile test to measure Young's modulus and breaking elongation of CEBH with different AC, as shown in Fig. 3d and Supplementary Fig. 4a. With the increase of AC, the Young's modulus of CEBH decreases gradually. When the AC is the same, the CEBH with higher water content results in a lower Young's modulus, when the water content reaches 72%, the Young's modulus decreases to 48 kPa, only 12% of the highest. The breaking elongation of CEBH decreased initially and then increased with the increase of AC. When the AC is relatively low, the elongation at break mainly depends on the properties of silicone rubber itself, and the layered structure leads to the decrease of elongation at break. However, when the poly (acrylic acid) was grafted through the silicone rubber, the increasing AC made CEBH a homogeneous hydrogel, thereby increasing the elongation at the break of CEBH. To explore the potential of CEBH in artificial skin, compression tests were also carried out using an oil hydraulic press, as shown the Fig. 3e and Supplementary Fig. 4b. The maximum bearing capacity of CEBH (8 mm × 8 mm × 3 mm) decreases first and then increases with the increase of AC, and reaches a maximum of 4000 kg

load without losing its shape recovery ability. However, the water content and swelling ratio of CEBH should be controlled, a relatively lower water content is needed to achieve better mechanical properties against pressure.

As shown in Fig. 3f and Supplementary Fig. 5a–g, we also performed puncture experiments on CEBH to evaluate its puncture resistance. Our findings indicate that CEBH can withstand up to 7.2 N of force, which is 6 times that of silicone rubber (The diameter of the needle is 1 mm). When the needle diameter is 10 mm, the puncture strength of CEBH is 96 N. During the experiment, obvious umbrella-shaped puncture deformation was observed, which is a sign of excellent puncture resistance. Besides, as shown in Supplementary Table 2, the anti-puncture force of CEBH in this work is much larger than the ordinary PAAm hydrogel, 8 folds of the SA-AAm double network hydrogel, comparable with the montmorillonite reinforced hydrogel, and 1/8 folds of hydrogel composites laminated with aramid fabric. Additionally, the friction coefficient of CEBH is also a critical factor for the artificial skin of the robotic arm. As shown in Supplementary Fig. 5h, i the friction coefficient of CEBH decreases slowly with the increase of water content, and can be adjustable between 0.36 and 1.3, which is similar to human skin, offering a candidate for cladding skin of robotic arms. In Fig. 3g and Supplementary Movies 3, 4, we demonstrate grabbing eggs with robotic arms coated with CEBH, both in air and water without damaging the eggshell. Moreover, the robotic finger coated with CEBH can flick the polystyrene foam beads instead of adhere to them (shown in Fig. 3h, Supplementary Fig. 6 and Supplementary Movie 5), which traditional cladding hydrogels cannot handle, showing similar anti-adhesion properties of the living skin reported by Shoji Takeuchi previously[45].

Therefore, CEBH is a hydrogel that closely emulates the properties of human skin, with excellent non-adhesion and non-brittleness endowing its high-pressure resistance and puncture resistance, which are two qualities of utmost importance in the mimicry of human skin. As shown in Supplementary Table 1, compared to the widely reported DN hydrogels and polyampholyte hydrogels, the CEBH has comparable compressive and tension modulus, and exhibits superior friction coefficient and puncture resistance. Besides, its anti-adhesion properties comparable with the living skin greatly benefit its application manipulator cladding layer, which is difficult for other double-network hydrogels. Overall, CEBH combines the advantages of existing elastomer and hydrogel materials, offering excellent comprehensive mechanical and surface properties that closely resemble real human skin.

## Stimulus-response, shape adaptability of CEBH and its applications

Shape adaptation of hydrogels is a crucial aspect of bionic applications, which often rely on the response to external stimuli[46,47], of which ion is a moderate and biocompatible trigger[48,49]. We demonstrated that CEBH, with a high content of carboxylic acid groups, can swell in an alkaline solution, fix its shape in $Ca^{2+}$ solution, and restore its original shape in weak acid solutions, which is illustrated in Fig. 4a. To test the ion response of CEBH, it was sequentially soaked in $Ca^{2+}$, dilute HCl, pure water, and $NH_4OH$ solution, and the reversible responses of CEBH in four solutions were shown in Fig. 4b. As shown in Fig. 4c, the CEBH film can be molded into predetermined shapes, such as a helix, S, or circular shape in $Ca^{2+}$ solution due to the interaction between $-COO^-$ and $Ca^{2+}$, and can recover its original shape in a dilute HCl solution, then gradually expand in water and ammonia solutions. Since the CEBH was composed of silicone rubber and PAA, which are both approved by the FDA for medical implants, these responsive properties can be beneficial for developing medical engineering materials, such as surgical repair and internal fixation materials. Soaking in $Ca^{2+}$ solution can also increase its fracture stress up to 5 MPa, with Young's modulus of 3.2 MPa. On the other hand, immersing the CEBH in an acid solution can increase the breaking elongation, but reduce its tensile

strength. Soaking $NH_3 \cdot H_2O$ significantly increases its swelling ratio, but also makes it fragile. The mechanic and swelling properties of CEBH immersed in different solutions are shown in Supplementary Fig. 7a, b. Given the non-toxic properties of these ions mentioned above, the responsive behaviors of CEBH provide excellent shape adaptability for its potential biomedical applications.

We conducted several application demonstrations utilizing the ion-responsive properties of CEBH. By converting part of a silicone rubber conduit into CEBH, the modified portion of the conduit is endowed with responsive features and shape adaptability triggered by different ions. A simple device that can control the liquid flow is depicted in Fig. 4d. In this device, a part of the conduit was converted into CEBH, and a small solid plastic ball was pre-set in this section. By changing the external ion environments between $Ca^{2+}$ solution and water, we successfully realized an ion-triggered switch of liquid flow through the expansion and shrinkage of the CEBH tube. As shown in and Supplementary Movie 6, when the silicone tube shrinks in $Ca^{2+}$ solution, blocking the pre-set ball and preventing the liquid from passing; when the external environment changes to water, the CEBH swells, increasing the diameter of the modified conduit and allowing the liquid to flow smoothly.

As another demonstration, the CEBH is used as a responsive bandage for shape-adaptive wound dressing (see Supplementary Movie 7). As illustrated in Fig. 4e, the composite hydrogel was first soaked in a weak alkaline solution and then applied to the wounded part in its expanded state. Calcium ion solutions were then sprayed to induce shrinkage, causing the bandage to conform to the contours of the wound and contract gently in just 60 s. When the bandage needs to be changed, a weak acid solution (0.2 M citric acid here) can be used to relieve the contraction, allowing the bandage to be removed easily. The entire process can be carried out in a moderate environment, without the need for any harmful reagents. A similar process can be employed as a surgical fixation method. In order to ensure that the use process does not cause harm to the human body, we record the $Ca^{2+}$ leakage of the calcified CEBH sheet (10 mm × 10 mm × 1 mm) immersed in 50 mL ultrapure water (shaken at speed of 110/min), and the ion chromatography results showed that the concentration of calcium ion in ultrapure water increased by only 27 µg/mL (Fig. 4f). Furthermore, we performed cytotoxicity experiments on calcified CEBH (MTT assay, Fig. 4g) and no significant difference was found when the leaching solution was added, proving that CEBH has good biocompatibility.

Full-thickness skin wound experiments on mice (BALB/c) are performed to determine the therapeutic effects of CEBH. Wounds with a diameter of 10 mm was produced on the dorsal skin of the experiment animals, and the healing rate of animals treated with Tegaderm™ film (control group) and CEBH (model group) was recorded. As shown in Fig. 4h, i and Supplementary Fig. 8a, the wound healing rates of control group (treated with Tegaderm™ film) and model group (treated with CEBH) were similar and there are no significant differences in the wound closure results, showing that the CEBH has similar effects with Tegaderm™ films on the wound healing of experiment mice. The Masson and HE staining images of wound sections on day 22 from both control and model groups are also shown in Supplementary Fig. 8b, showing the CEBH didn't introduce additional skin fibrosis or inflammatory cells than control group. Therefore, the CEBH promoted wound healing similarly with Tegaderm™ films, and showed good biosafety properties. The contractile force of the CEBH tube can provide a solid connection, affording a pulling force as large as 16 N, as demonstrated in Fig. 4j and Supplementary Fig. 7c. Consequently, the shape adaptability of CEBH makes it an ideal candidate for solid fixation in surgeries.

## Discussion

In summary, we proposed a straightforward and versatile method for synthesizing CEBH using a penetrating-derived radiation grafting

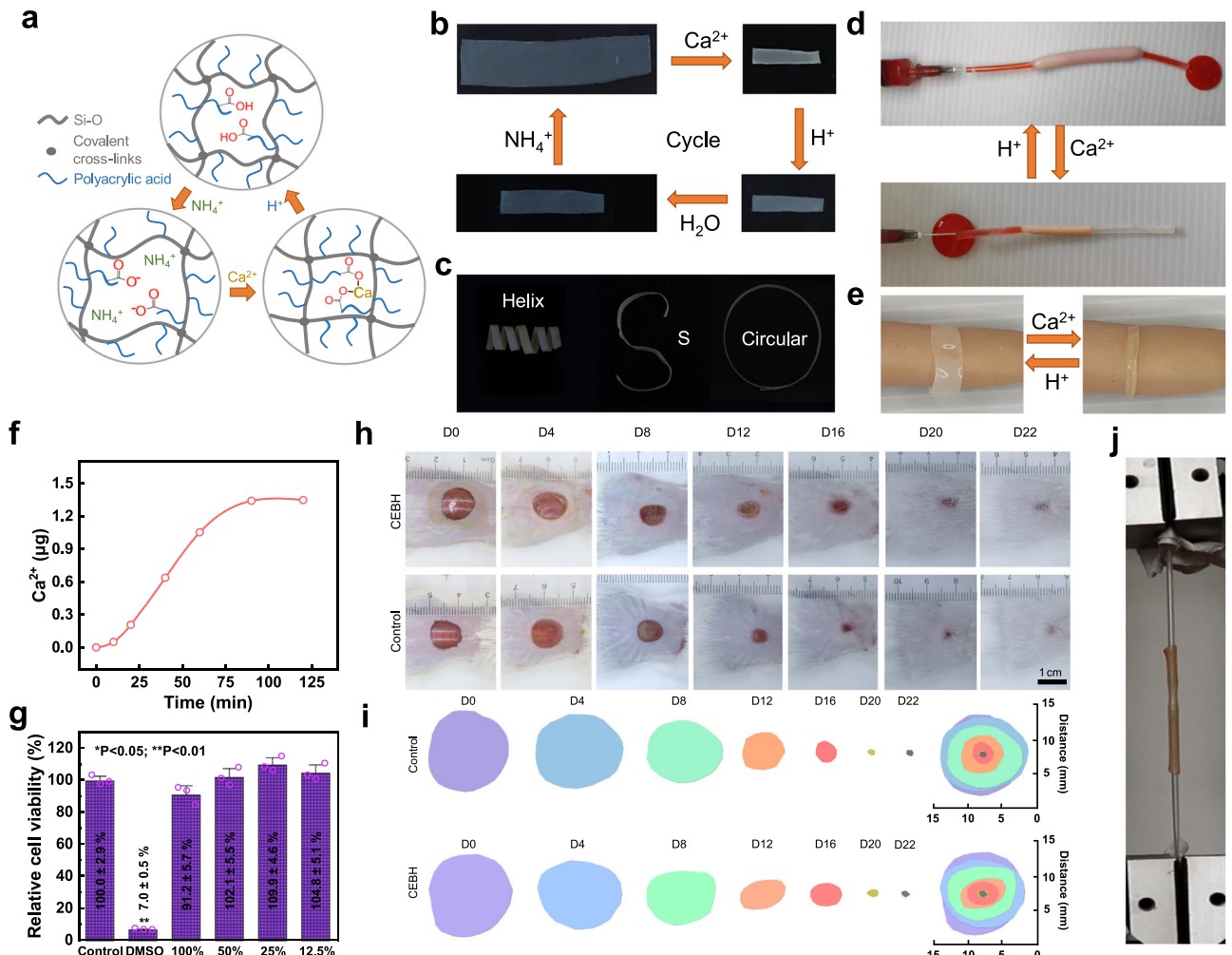

**Fig. 4 | Shape adaptability of CEBH. a** The conceptional illustration of reversible responses of CEBH. **b** The ion response process of CEBH in the ammonia, Ca²⁺, HCl solution, and water. **c** 3D-shape adaptability of CEBH films via immersing in Ca²⁺ solution. **d** Liquid flow control and **e** shape-adaptive wound dressing using ion-responsive CEBH on a highly simulated silicone skeleton hand model. (The acid environment is supplied with 0.2 M citric acid, and the Ca²⁺ solution is used to trigger the shrinkage of CEBH. The liquid flowing in the tube was dyed using 0.2% of eosin Y.) **f** Calcium ion leakage of calcified CEBH in 50 mL ultrapure water. **g** The viability of L929 cells co-cultured with extraction solution of the calcified CEBH for 24 h (MTT assay). Data are presented as mean values ± SD ($n = 3$ independent cell experiments). Two-tailed t-tests were performed between experiment groups and the control group, and $p$ values were <0.0001, 0.1145, 0.9653, 0.0701, and 0.5679 for 20% DMSO, and extraction solutions with concentrations of 100%, 50%, 25%, and 12.5%. **h** Representative photographs and **i** traces of wound closure of skin wounds treated with Tegaderm film™ (control group), CEBH (model group) on days 0, 4, 8, 12, 16, 20 and 22. **j** Tensile test of two tubes connected by shrinking of CEBH in Ca²⁺ solution. In (**a**–**h**), the concentration of ammonia was 0.2%, CaCl₂ was 0.1 M, and the AC of CEBH was 62%. In (**a**–**d**), the concentration of HCl was 1%.

technique by simply irradiating elastomers immersed in an AAc monomer solution. The CEBH we reported preserves the elastic chain network of the original elastomers, and changes into hydrophilic and swells in water after PAA chains were introduced at the molecular chain scale. Combining the advantages of existing elastomer and hydrogel materials, the resulting CEBH possesses excellent comprehensive mechanical and surface properties comparable to real human skin. In addition, CEBH exhibits ion-responsive behavior, enabling CEBH shape adaptability, which is critical in various bionic applications. These unique properties make CEBH suitable for use as artificial skin, fluid flow controller, as well as wound dressing layer. Furthermore, the synthetic strategy is broadly applicable to a variety of common elastomers, including polydimethylsiloxane, natural latex, fluorine gum, polyurethane, SEBS, VHB, and others.

## Methods

### Materials

Acrylic acid (99%, AR), isopropanol (99.5%, AR), sodium hydroxide (96%, AR), eosin Y (80%, AR), DMSO (99.7%, AR), MTT (98%) were purchased from Macklin Technology Co., Ltd. Ammonium ferrous sulfate hexahydrate (99.5%, AR), calcium chloride (96%, AR) hydrochloric acid (38%, AR), citric acid (99%, AR), toluene (AR), cupric chloride (99%) were purchased from Sinopharm Chemical Reagent Co., Ltd. Fetal bovine serum was purchased from Procell. MEM medium (10-010-CVRC) was purchased from Corning. PBS (P1022) was purchased from Solarbio. Pancreatic enzyme (BL512A) was purchased from Biosharp. L929 cell line (mouse fibroblast, cat. GNM28) was obtained from and have been authenticated by the Cell Bank of the Chinese Academy of Sciences (Shanghai, China). Isoflurane (20220802) was purchased from RWD Co., Ltd. Normal saline (R13B22091201) was purchased from the Shandong Qidu pharmaceutical industry. Tegaderm™ transparent wound dressing was purchased from 3 M. All additional chemical reagents were of analytical grade and used without further purification.

### Devices and characterization

Scanning electron microscopy images were recorded using a field-emission scanning electron microscope (Hitachi SU8220, Japan) to

determine the morphologies of CEBH. The photographs and movies were taken with a digital camera (Nikon D3100). The radiation experiments were carried out at the Radiation Center of University of Science and Technology of China and the Radiation Center of Nanjing University of Aeronautics and Astronautics. Strain testing is carried out using HZ-1004B tensile test machine (Hengzhun Instrument Technology Co., Ltd). Differential scanning calorimetry is carried out on a NETZSCH 200F3. Fourier transform infrared spectra are recorded using WQF-510 (Beijing Rayleigh). Small-angle neutron scattering was carried out on the Suanni small-angle neutron spectrometer at CMRR (China Mianyang Research Reactor).

Instruments used in animal experiments: ivc cages (Suhang Science and Technology Co., Ltd, G4); Digital camera (Canon, EOSM6); Small animal anesthesia machine (RWD, H1670401-200L); Veterinary oxygen generator (RWD, SN399884).

## Preparation of composite hydrogel

The elastomers were washed with isopropyl alcohol and pure water and dried in an oven. Measure and record the quality after drying. 25 wt% acrylic acid and 0.25–0.75 wt% Mohr's salt were dissolved in deionized water and stirred evenly to obtain a mixed solution. The mixed solution and elastomer were added into a glass vial, bubbled with argon to repel oxygen, and then sealed. The sealed vial was irradiated with $^{60}$Co with absorbed doses of 10, 30, 50, and 70 kGy, respectively (with a dose rate of 0.45 kGy·h$^{-1}$). After radiation processing, the elastomers were cleaned and extracted with a Soxhlet extractor, and dried to a constant weight before weighting. The formulas for acrylic acid content (AC) (1) and swelling ratio ($S_a$) (2) are as follows:

$$AC(\%) = 100 \times (W_1 - W_0)/W_1 \tag{1}$$

$$S_a(\%) = 100 \times (W_2 - W_1)/W_1 \tag{2}$$

where $W_0$ is the mass of elastomer before grafting; $W_1$ is the mass after grafting; and $W_2$ is the mass after complete water absorption and expansion.

## Mechanical testing

The CEBH film was cut into dumbbells before tensile tests. The tensile rate is kept at 20 mm min$^{-1}$, and the Young's modulus is determined by the slope of the stress-strain curve. In the compressive tests, the samples with a size of 8 mm × 8 mm × 3 mm were used, and the compression rate was kept at 1 mm/min. The compression stress was determined from the slope of the stress-strain curve. The puncture test was carried out using a circular plate (diameter: 70 mm) and a CEBH film was punched by a cylindrical indenter (diameter: 1 mm and 10 mm) at a speed of 20 mm min$^{-1}$. The friction coefficient is measured by the friction angle method and an average was calculated from 3 parallel tests.

## Ion response experiment of CEBH

0.1 M CaCl$_2$, 0.25% NH$_3$·H$_2$O, 1% HCl solution and deionized water were used in the experiments. CEBH film (thickness of 0.3 mm) was sequentially immersed in pure water, NH$_3$·H$_2$O, CaCl$_2$ solution, and HCl for ~60 s.

## Calcium ion release experiment of CEBH after calcification

The alkaline CEBH was first immersed in a 0.1 M calcium ion solution until its volume did not change. Then, the calcified CEBH sheet (10 mm × 10 mm × 1 mm) was immersed in 50 mL ultrapure water and shaken at 110 speeds on a shaker. 2 mL of solution was taken and characterized using ion chromatography (IC1600, Qingdao Yiyi Instrument Co., Ltd.).

## Cytotoxicity test of calcified CEBH (MTT assay)

The L929 cells were cultured in MEM medium containing 10% FBS in a humidified atmosphere with 5% CO$_2$ at 37 °C. The viability of L929 cells was studied using the standard MTT assay. Cells were seeded on 96-well plates (8 × 10$^3$ cells per well) and incubated for 24 h before the next experiments. The samples were cut into 2 cm × 2 cm and then extracted in 3 mL complete medium (1.25 cm$^2$/mL) for 24 h. Different concentrations of extraction solution (100%, 50%, 25%, 12.5%, and 0% as well as 20% DMSO) were added to the medium and incubated with the cells for 24 h and then MTT assay was conducted. One-way ANOVA with Tukey's multiple comparisons was used for comparisons between control and model groups, and two-tailed Student's t-test was used and $P < 0.05$ were considered significant.

## In vivo wound healing experiment of CEBH

After 7 days of adaptive feeding, 6 female mice (BALB/c, 8 weeks old) were randomly assigned to the control group and the model group and a full-thickness cutaneous wound with a diameter of 10 mm was created on the dorsal of all 6 mice. The injury was treated with Tegaderm™ film (control group, $n = 3$) and the CEBH film (model group, $n = 3$). Subsequently, mice were kept individually with food and water in a specific pathogen-free (SPF) animal room. The wound healing status was captured by a digital camera at every two days, and wound closure degree can be calculated based on the traces of wound areas. One-way ANOVA with Tukey's multiple comparisons was used for comparisons between control and model groups, and two-tailed Student's t-test was used and $P < 0.05$ were considered significant. On day 22, the wound site and adjacent normal skins were harvested and then sectioned for H&E and Masson's trichrome staining. All animal experiments were approved by the Animal Ethics Committee of Yanxuan Biotechnology (Hangzhou) Co., Ltd. (Approval No. YXSW2309269698).

## Reporting summary

Further information on research design is available in the Nature Portfolio Reporting Summary linked to this article.

# Data availability

The authors declare that the data supporting the findings of this study are available within the article and its Supplementary Information. Source data are provided with this paper (https://doi.org/10.6084/m9.figshare.24041514). Additional data are available from the corresponding author upon request.

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

## Acknowledgements

This work was supported by the National Natural Science Foundation of China (22006067, 11975122, 21906083, 12004180), the National Key R&D Program of China (2022YFB1902900), Scientific and Technological Innovation Special Fund for Carbon Peak and Carbon Neutrality of Jiangsu Province (BK20220026), Fundamental Research Funds for the Central Universities (NT2022020). The authors would like to thank China Mianyang Research Reactor for collecting Neutron Scattering Data and An Yusen from Shiyanjia Lab (www.shiyanjia.com) for the assistance of animal wound healing experiment.

## Author contributions

Conceptualization: W.Y. and M.J.; methodology: W.Y. and T.Y.; investigation: T.Y., W.Y., W.Z., C.S., L.D., and J.Z.; visualization: W.Y., T.Y., and Z.Y.; supervision: M.J. and W.Y.; writing—original draft: T.Y., W.Y., and C.C.; writing—review & editing: M.J. and W.Y.; funding acquisition: W.Y. and M.J.

## Competing interests

The authors declare no competing interests.
