## [Peer Review File · Nature Communications]

Connective Tissue Inspired Elastomer-based Hydrogel for Artificial Skin via Radiation-induced Penetrating PolymerizationReviewers' Comments:

Reviewer #1:

Remarks to the Author:

The work demonstrates an elastomer-hydrogel composite using gamma radiation grafting polymerization. The work is interesting but in the current form it has some major issues. The mechanism of grafting is not explained clearly and needs more proof. Further, the characterization such as mechanical is not extensively performed and again lacks explanation for the observed behavior. The application part is not convincing. Overall, the manuscript is not a good fit for this journal in the current form.

1. The polymer grafting method discussed by authors here to form elastomer-hydrogel hybrid has been reported earlier in similar hydrogel-based systems. In a report by Huang et al in 2007 (<https://onlinelibrary.wiley.com/doi/10.1002/adma.200602533>), gamma radiation result in grafting of acrylic acid monomers onto microspheres of styrene and butyl acetate. The hybrid hydrogel report here had a strength of ~ 10 MPa. The gels could also swell to 900 times their dry weight suggesting longer chain between two crosslinks. This raises questions on the efficiency of the strategy reported in this work. The authors should therefore justify why and how their strategy is better.

2. What do authors mean by 'Moore's salt'? Is it confused for 'Mohr's salt'(Ammonium iron(II) sulfate)?

3. The salt is used as a radical scavenger to prevent polymerization in the solution. However, gamma radiation is known to generate free radicals in water (H^+ and OH^-). How does this affect the polymerization of acrylic acid during the radiation exposure?

4. The FTIR peaks at 1702 and 2962 cm^{-1} show that acrylic acid is present, but it does not prove that it is grafted onto to the elastomer. Was there any peak shift observed for lower wavelengths as the radiation dosage is increased? At this point it would be advisable to define what authors mean by grafting. Grafting polymerization is generally associated with covalent linkage of monomers as a branched network to the main chain.

For the DSC curve, the weight loss pattern for irradiated samples should be similar, but in Figure 1c, 50 and 70kGy almost follow similar trend but lower dosages have different weight loss patterns. What is the reason for this? One understanding could be presence of unreacted acrylic acid for lower radiation dosage. What do authors think about this?

5. Figure S3: It would have been better if the swelling ratios for toluene and water were studied for similar weight fraction of AC.

6. Figure 2a is confusing. It is not very clear as to what authors want to deduce. It is claimed that swelling ratio in toluene decreases as AC concentration is increased. However, from the images it is not very evident. Having a quantitative analysis in terms of swelling ratio comparison will be helpful.

7. Figure 2b: How was the AC concentration measured inside elastomers of different hardness?

8. Page 9: Authors mention about increase in contractile force as AC content increased. Is there any quantifiable method based on which this statement is made or it is just a property of PAA to induce more contraction in general.

Reviewer #2:

Remarks to the Author:

In this manuscript, the authors reported a one-step radiation grafting method for preparing connective-tissue-inspired elastomer/hydrogel hybrids (CEHH) with commercial crosslinking elastomers. The as-obtained CEHH possesses both the advantages of elastomers and hydrogel, presenting mechanical and surface properties comparable to human skins. Besides, CEHH shows ion-responsive behavior, making it suitable for several bionic applications. Here I suggest a minor revision before it can be accepted for publication. Some specific key points are as follows:

1. In Figure. S2a and c, the AC values are quite different at the same experimental condition. For

example, whether the AC is 20% or 0.2% at 30kGy? Please check the results carefully.

2. The authors used water and toluene absorption tests to confirm the hydrophilicity of CEHH. Here contact angle tests are recommended for further estimation.

3. According to the reference literature and the discussion of the "grafting front" mechanism in this manuscript, the grafting process should occur from the surface to the inside part. However, the sentence "Therefore, the macroscopic-scale elastomers can be grafted from the inside out if the processing process is well controlled" make me confused, whether it is from the outside or inside?

4. There are several description mistakes and misnomers throughout the manuscript. For example, in the introduction part, about the description of the challenges of fabricating heterogeneous hydrogel, the authors claimed that the precursors are "hydrophilic and insoluble in water" and listed several hydrophobic materials, which makes the sentence misleading. Please check it carefully.

Reviewer #3:

Remarks to the Author:

This article presents the development of a strategy to convert elastomeric backbones into hydrogels via radiation-induced penetrating polymerization with applications in artificial skin. While the polymerization strategy is interesting, the paper fails to provide a clear explanation of the novelty of the proposed material in comparison to existing tough hydrogels. Detailed comments required for clarification are provided below.

1. The authors claim that the connective-tissue-inspired elastomer/hydrogel hybrids (CEHH) can be utilized to achieve realistic human skin, while previously reported hydrogels cannot do so. It is unclear which properties the CEHH excels at compared to previous hydrogels. The paper states compression and puncture load capabilities, but no comparison study was conducted. A systematic comparison of the key material properties (compressive modulus, puncture load capabilities, Young's Modulus, stretchability, friction coefficient) in comparison to previous works should be provided in a table.

2. An application proposed by the authors is an artificial-skin material comparable to real skin. Please demonstrate that the mechanical properties (Young's modulus, compressive modulus, puncture load capabilities) of the CEHH can be tuned to match the mechanical properties of various types of skin layers (epidermis, dermis, stratum corneum, hypodermis).

3. The authors further claim that the CEHH is similar to connective tissue. However, there are no results that demonstrate the anatomical or structural similarity with connective tissue and no indication of what type of connective tissue. All contents related to inspiration from connective tissue should be removed.

4. Please provide video comparison of CEHH with traditional cladding hydrogels in Supporting Movie 5 to demonstrate that the polystyrene foam beads stick to traditional cladding hydrogels.

5. The potential of the radiation-induced penetrating polymerization to create a wide variety of hydrogels is very interesting. However, the paper only attempts 3 types of elastomers: PDMS, natural latex, and VHB. Please demonstrate the versatility of this technique by testing additional elastomers such as polyurethane, styrene-ethylene-butylene-styrene, cyclic olefin copolymer elastomers, and fluoroelastomer.

6. The paper claims biocompatibility of the CEHH. However, no results have tested for biocompatibility. Please report cell viability results.

7. The shape adaptability of CEHH for potential biomedical applications is demonstrated with ionic stimuli such as Ca^{2+} or citric acid. However, the concentrations utilized in the experiments (0.1M CaCl_2 acts as a body tissue irritant and is slightly toxic during ingestion

[https://www.mccsd.net/cms/lib/NY02208580/Centricity/Shared/Material Safety Data Sheets _MSDS_/MSDS Sheets_Calcium_Chloride_solution_0_1M_148_00.pdf](https://www.mccsd.net/cms/lib/NY02208580/Centricity/Shared/Material%20Safety%20Data%20Sheets/_MSDS_/MSDS%20Sheets_Calcium_Chloride_solution_0_1M_148_00.pdf)) are much higher than physiologically relevant concentrations ($\sim 2\text{mM}$ calcium). Please demonstrate the shape adaptability at physiologically relevant concentrations.

8. The name elastomer/hydrogel hybrids is very confusing as it leads the reader to think of conventional elastomer/hydrogel bilayer hybrids. Renaming the material to reflect that the elastomer backbone itself becomes a hydrogel would provide clarification.

Response to Reviewers' Comments

Response to the comments of Reviewer #1

Comment No.1: The polymer grafting method discussed by authors here to form elastomer-hydrogel hybrid has been reported earlier in similar hydrogel-based systems. In a report by Huang et al in 2007 (<https://onlinelibrary.wiley.com/doi/10.1002/adma.200602533>), gamma radiation result in grafting of acrylic acid monomers onto microspheres of styrene and butyl acetate. The hybrid hydrogel report here had a strength of ~ 10MPa. The gels could also swell to 900 times their dry weight suggesting longer chain between two crosslinks. This raises questions on the efficiency of the strategy reported in this work. The authors should therefore justify why and how their strategy is better.

Response: Thanks for the reviewer's comments.

There is a conceptual difference between the high-performance microspheres hydrogel proposed by Huang et al and our design in the current manuscript. First, the robust hydrogel they proposed is based on acrylic acid grafting on the surface of microspheres and form a heterogeneous network with microspheres as crosslinking nodes. In contrast, our system begins with a highly crosslinked ubiquitous elastomer. The hydrogels are achieved by one step irradiation grafting of acrylic acid within the network of the elastomer. Therefore, the preparation of our hydrogel is broadly applicable to elastomers materials and more facile, which has great potentials in a large scale manufacture. Secondly, it is emphasized that the network from the original elastomer that provides the elasticity, instead of the poly(acrylic acid) PAA chains. The microspheres in the hydrogel reported by Huang et al are visible, while our material showed a homogenous structure at the micrometer scale. The network composed of a hydrophobic elastic network and hydrophilic graft chains makes it swell in water and avoid the brittleness of traditional hydrogels. As shown in the following table, the Compressive and tension modulus can be as well as the widely reported DN hydrogels and polyampholyte hydrogels, and exhibits superior friction coefficient and puncture resistance compared to double network hydrogels. Besides, as we showed in the supporting movie, the foam ball can be easily bounced away by the manipulator with our hydrogel as cladding layer, which is difficult for other double-network hydrogels.

Comparison of mechanical properties of CEBH, DN hydrogel, Human skin.

Hydrogel	Yong's modulus (MPa)	Friction Coefficient	Compressive Resistance (MPa)	Puncture Resistance (MPa)
CEBH	0.048-3.2	0.36-1.3	5.7-620	1.53-9.17*
BC-PVA-PAMPS	155-227	0.06	17.3-23	-- ^[1]
SA-AAm	--	--	--	0.0048-0.038** ^[2]
Agar-PAAm	0.08	--	38	-- ^[3]
PVDT-PEGDA	0.12	--	6	-- ^[4]
HA-SS-PEG	0.01-0.05	--	0.08-0.32	-- ^[5]
Agarose hydrogel	--	0.005-0.09	1	-- ^[6]
Highly entangled	35	0.0067	--	-- ^[7]

PAAm hydrogel				
PAAN	0.006-0.07	--	0.54-8.53	--[8]
Chitosan-gelatin- phytate	0.03-2.47	--	35.7-64	--[9]
Human skin	0.1-2	0.4-0.8	0.3	3.18

* The diameter of the indenter is 0.5 mm. **The diameter of the indenter is 10 mm.

Comment No.2: What do authors mean by ‘Moore’s salt’? Is it confused for ‘Mohr’s salt’ (Ammonium iron (II) sulfate)?

Response: We’re sorry for this mistake. We have corrected it as Mohr’s salt in the manuscript.

Comment No.3: The salt is used as a radical scavenger to prevent polymerization in the solution. However, gamma radiation is known to generate free radicals in water (H⁺ and OH⁻). How does this affect the polymerization of acrylic acid during the radiation exposure?

Response: The homopolymerization of acrylic acid make the solution viscosity and reduce the grafting efficiency. To realize penetrating grafting polymerization, the competing homopolymerization should be avoid. The simplest strategy to do so is to use the scavenger to quench those radicals in solutions which can initiate homopolymerization. For instance, Cu²⁺, Fe²⁺ ions have been often applied as inhibitors in the radiation induced grafting systems since the 1960s. As reported by previous literature (Journal of Polymer Science A, 1969, 7(6): 1379-1384. <https://doi.org/10.1002/pol.1969.150070601>), ferric ions can inhibit both the graft polymerization and the homogenous polymerization as follows:

after the polymerization reaction was initiated by the free radicals by the radiolysis of water.

As the reviewer point out, water radiolysis produces reactive species including OH, and H radicals. It is known that OH radicals represent the dominating species for AA polymerization in bulk phase. Once scavenger like ferric ions is present, it could quench the initiating radicals. Fortunately, this effect works differently in the solution (where photopolymerization takes place) and at the surface and inside the base of the polymer bases (where grafting polymerization takes place). The “selective inhibition of homo-polymerization” is recognized by the difficulty of Fe²⁺/Fe³⁺ diffusion to the surface of the polymer base, and the complex of the Fe²⁺/Fe³⁺ with PAA ((Journal of Polymer Science A, 1969, 7(6): 1379-1384. <https://doi.org/10.1002/pol.1969.150070601>; International Journal of Radiation Applications and Instrumentation C, 1989, 33(1): 51-60. [https://doi.org/10.1016/1359-0197\(89\)90094-5](https://doi.org/10.1016/1359-0197(89)90094-5)). As a result, the small amount of salt could benefit the grafting reactions, and it has been shown by our comparison experiments. These discussions have been added to the revised manuscript.

Comment No.4: The FTIR peaks at 1702 and 2962 cm⁻¹ show that acrylic acid is present, but it does not prove that it is grafted onto to the elastomer. Was there any peak shift observed for lower wavelengths as the radiation dosage is increased? At this point it would be advisable to define what

authors mean by grafting. Grafting polymerization is generally associated with covalent linkage of monomers as a branched network to the main chain.

For the DSC curve, the weight loss pattern for irradiated samples should be similar, but in Figure 1c, 50 and 70kGy almost follow similar trend but lower dosages have different weight loss patterns. What is the reason for this? One understanding could be presence of unreacted acrylic acid for lower radiation dosage. What do authors think about this?

Response: Thanks for the reviewer's comments. As the reviewer points out, the FT-IR spectra can only prove that there are PAA content exists in the hydrogel, which cannot be proved to be grafted or mixed in the material. No obvious peak shift is observed for lower wavelengths as the radiation dosage is increased. Before further characterization, the hydrogel is extracted with water for days to remove the presence of unreacted monomers and photopolymerization products. To obtain direct evidence of the grafting structure, the solid-state ^{13}C -NMR of the original PDMS and hydrogel are provided in the revised manuscript. As shown in the ^{13}C -NMR spectroscopy of the dried hydrogel, the peak at chemical shift of 8.80 clearly shows the presence of the carbon atom of AAc grafted -O-Si(CH₂-g-AAc)- was shifted to a higher field, which proves the presence of covalent linkage and the successful grafting of AAC as a branch onto the PDMS chain network via covalent bonds. These newly obtained data have been discussed in the revised manuscript.

^{13}C -NMR spectra of silicone rubber and CEBH

The weight loss process read from Thermogravimetric differential curve (DTG) was attributed by previous literature (Journal of Applied Polymer Science, 2007, 105(6): 3220-3227. <https://doi.org/10.1002/app.26267>). The first step (before 200°C) is attributed to the loss of water; the second step (235-329°C), is attributed to the formation of PAA anhydride; the third step (331-512°C) is attributed to the degradation of the corresponding PAA anhydride; the final step (up to 693 °C) is attributed to the thermal degradation of backbone. As the content of PAA increases, the

bound water will increase, and the self-association of carboxyl groups will increase, which is beneficial to the second and third weight loss steps. These discussions have been added in the revised manuscript.

Besides, it should be noted that the silicone rubber and PAA are highly incompatible materials, there will be obvious phase separation no matter how “homogenously” they are mixed, and that’s why we performed DSC testing, which is to prove the phase status in the material. As shown in the DSC curve, the two-phase transition processes of the two compositions at last became one as the degree of grafting increased.

Comment No.5: Figure S3: It would have been better if the swelling ratios for toluene and water were studied for similar weight fraction of AC.

Response: Thanks for the reviewer's suggestion. We have revised this to present the swelling ratio in toluene in a similar manner to make it visual and direct to compare with the water swelling ratio.

Comment No.6: Figure 2a is confusing. It is not very clear as to what authors want to deduce. It is claimed that swelling ratio in toluene decreases as AC concentration is increased. However, from the images it is not very evident. Having a quantitative analysis in terms of swelling ratio comparison will be helpful.

Response: Thanks for the reviewer's comments. We marked the grafting front in the figure. With the increase of AC, the unmodified silicone rubber in the middle gradually becomes thinner until it completely disappears. The approximate value of the unmodified thickness is marked in the following figure.

(a) Digital images of CEBH with different acrylic acid contents after being swollen in 1 g/mL Cu²⁺ solution (left) and toluene (right). (b) Thickness of unmodified layers of gels with different AC immersed in water and toluene respectively.

Comment No.7: Figure 2b: How was the AC concentration measured inside elastomers of different hardness?

Response: Firstly, the excess acrylic acid and iron ions in the sample were removed by Soxhlet extractor, and then the AC was obtained by the weighing method. The formula is as follows (7):

$$AC (\%) = 100 * (w_1 - w_0) / w_1 \quad (7)$$

w₀ is the dry weight of the sample before modification, and w₁ is the dry weight of the sample after modification.

Comment No.8: Page 9: Authors mention about increase in contractile force as AC content increased. Is there any quantifiable method based on which this statement is made or it is just a property of PAA to induce more contraction in general.

Response: Thanks for the reviewer's comments. The increment of contractile force will increase as the swelling ratio increases in elastomer or gels, which has been proved by Paul J. FLORY (Polymer Journal, 1985, 17(1), 1, <https://doi.org/10.1295/polymj.17.1>). The increment of swelling increases the contractile forces delivered by the networks of Gaussian chains in the elastomer, which is proportional to the displacement lengths of the Gaussian chains (proportional to $(V/V_0)^{1/3}$).

The contractile force of the swollen PDMS in toluene reported by Flory (Polymer Journal, 1985, 17(1), 1)

The effect of contractile force on the grafting was verified by the grafting of silicone rubber with different crosslinking degrees, which means they can provide different contractile force when swollen. As shown in Figure.2b and Figure.S3c, the degree of grafting sharply decreases with the crosslinking degree of silicone rubber while other conditions remain the same, proving that the shrinkage force caused by the denser Gaussian chain network is a key factor that prevents the further increasing of the DG, which can be concluded to the larger shrinkage force that hinders the grafting of acrylic acid. Unfortunately, there is no good way to quantify this shrinkage force in experiment. This discussion has been added in the revised manuscript.

Response to the comments of Reviewer #2

Comment No.1: In Figure. S2a and c, the AC values are quite different at the same experimental condition. For example, whether the AC is 20% or 0.2% at 30kGy? Please check the results carefully.

Response: Thanks for the reviewer's suggestion. This is a unit conversion error that mentions the use of percentage, and we have corrected it in the revised manuscript.

Comment No.2: The authors used water and toluene absorption tests to confirm the hydrophilicity of CEHH. Here contact angle tests are recommended for further estimation.

Response: Thanks to the reviewers for their comments. It can be seen from the figure below that with the increase of AC, the contact angle decreases, which can verify the increase of hydrophilicity. The result has been added in the revised manuscript.

Water contact angle of modified silica gel with different AC

(0%, 43%, 62% respectively from left to right).

Comment No.3: According to the reference literature and the discussion of the “grafting front” mechanism in this manuscript, the grafting process should occur from the surface to the inside part. However, the sentence “Therefore, the macroscopic-scale elastomers can be grafted from the inside out if the processing process is well controlled” make me confused, whether it is from the outside or inside?

Response: Thanks to the reviewers for their comments. We changed the sentence as “Therefore, the macroscopic-scale elastomers can be grafted from the outside to the inside if the processing process is well controlled.”

Comment No.4: There are several description mistakes and misnomers throughout the manuscript. For example, in the introduction part, about the description of the challenges of fabricating heterogeneous hydrogel, the authors claimed that the precursors are “hydrophilic and insoluble in water” and listed several hydrophobic materials, which makes the sentence misleading. Please check it carefully.

Response: Thanks to the reviewers for their comments. The word hydrophilic should be hydrophobic. We have carefully checked and revised the misleading expression throughout the manuscript.

Response to the comments of Reviewer #3

Comment No.1: The authors claim that the connective-tissue-inspired elastomer/hydrogel hybrids (CEHH) can be utilized to achieve realistic human skin, while previously reported hydrogels cannot do so. It is unclear which properties the CEHH excels at compared to previous hydrogels. The paper states compression and puncture load capabilities, but no comparison study was conducted. A systematic comparison of the key material properties (compressive modulus, puncture load capabilities, Young’s Modulus, stretchability, friction coefficient) in comparison to previous works should be provided in a table.

Response: Thanks to the reviewers for their comments. The CEBH is a hydrogel with comparable properties to those of skin, and with excellent non-adhesion and non-brittleness endows it high pressure resistance and puncture resistance, which are critical in the artificial skin. As shown in the following table, the Compressive and tension modulus can be as well as the widely reported DN hydrogels and polyampholyte hydrogels, and exhibits superior friction coefficient and puncture resistance compared to double network hydrogels. Besides, as we showed in the supporting movie, the foam ball can be easily bounced away by the manipulator with our hydrogel as cladding layer, which is difficult for other double-network hydrogels. This discussion has been added in the revised manuscript.

Comparison of mechanical properties of CEBH, DN hydrogel, Human skin.

Hydrogel	Yong’s modulus (MPa)	Friction Coefficient	Compressive Resistance (MPa)	Puncture Resistance (MPa)
CEBH	0.048-3.2	0.36-1.3	5.7-620	1.53-9.17*
BC-PVA-PAMPS	155-227	0.06	17.3-23	-- ^[1]
SA-AAm	--	--	--	0.0048-0.038** ^[2]
Agar-PAAm	0.08	--	38	-- ^[3]

PVDT-PEGDA	0.12	--	6	--[4]
HA-SS-PEG	0.01-0.05	--	0.08-0.32	--[5]
Agarose hydrogel	--	0.005-0.09	1	--[6]
Highly entangled PAAm hydrogel	35	0.0067	--	--[7]
PAAN	0.006-0.07	--	0.54-8.53	--[8]
Chitosan-gelatin- phytate	0.03-2.47	--	35.7-64	--[9]
Human skin	0.1-2	0.4-0.8	0.3	3.18

* The diameter of the indenter is 0.5 mm. **The diameter of the indenter is 10 mm.

Comment No.2: An application proposed by the authors is an artificial-skin material comparable to real skin. Please demonstrate that the mechanical properties (Young's modulus, compressive modulus, puncture load capabilities) of the CEHH can be tuned to match the mechanical properties of various types of skin layers (epidermis, dermis, stratum corneum, hypodermis).

Response: Thanks to the reviewers for their comments. The skin is mainly divided into three layers, namely the epidermis, dermis, and subcutaneous tissue. The stratum corneum belongs to the epidermis, in the outermost layer of the epidermis. CEHH is developed to serve as the cladding layer of the robot, so the positioning of the material is to mimic the overall skin. The Young's modulus of the skin is 0.1-2 MPa (G. A. Holzapfel, *Mechanics of Biological Tissue*, Springer, Berlin 2006), the friction coefficient is about 0.4-0.8 (Zhang M et al. in 1999, doi: 10.3109/03093649909071625.), and the puncture resistance strength is 3.183 MPa (Henry S et al. in 1998, doi: 10.1021/js980042+.), which are all within the adjustable range of CEHH that developed in present study. The Young's modulus is 0.048-3.2 MPa, the friction coefficient is 0.36-1.3, and the puncture resistance strength is 1.2-7.2 N (1.53-9.17 MPa), so it is regarded as a comparable material to human skin.

Comment No.3: The authors further claim that the CEHH is similar to connective tissue. However, there are no results that demonstrate the anatomical or structural similarity with connective tissue and no indication of what type of connective tissue. All contents related to inspiration from connective tissue should be removed.

Response: Thanks for the suggestion. The concept of connective tissue in this article is narrowly defined, mainly referring to the connective tissue proper (connective tissues except bone, cartilage, blood, and lymph), not including specialized connective tissue. We admit that judgment on the question that if the material is connective-tissue-like is subjective. This problem is likely due to the challenges of acquiring sufficient knowledge of connective tissues constitutes and understanding how they work. Researchers around the world have made considerable progress, but it still remains a synthetic challenge. From the existing literature, the functionality of connective tissues is attributed to the soft biological structure consisting of elastic collagen fibers, water-rich ground substances, and cells that generate the two components above. The elastic collagen fibers give them high elasticity and tough mechanical properties, while the ground substance provides hydrophilic properties. From this basis, we attempted to fabricate a hydrogel with an elastic network (the chain network of the original elastomer) and water-rich substrate (grafted PAA chains and the water swelling them) composed of different polymer chains respectively. This is much different from traditional hydrogels, which consist of one or more networks (DN hydrogel) and these networks attribute elasticity and hydrophilicity at the same time. In our humble opinion, we think our hydrogel successfully imitates these parts that compose the connective tissues, and

the imitation endows the hydrogel similar properties as connective tissues. We agree with the reviewer that the hydrogel materials are not completely connective tissues-like materials. However, the material conception is stemmed from connective tissues inspiration. We wish that our preliminary work and radiation strategy may lead to more advances in artificial skin.

Comment No.4: Please provide video comparison of CEHH with traditional cladding hydrogels in Supporting Movie 5 to demonstrate that the polystyrene foam beads stick to traditional cladding hydrogels.

Response: Thanks to the reviewers for their comments. We provided a video comparison of CEBH with traditional cladding hydrogels to demonstrate that the polystyrene foam beads stick to traditional cladding hydrogels.

Comment No.5: The potential of the radiation-induced penetrating polymerization to create a wide variety of hydrogels is very interesting. However, the paper only attempts 3 types of elastomers: PDMS, natural latex, and VHB. Please demonstrate the versatility of this technique by testing additional elastomers such as polyurethane, styrene-ethylene-butylene-styrene, cyclic olefin copolymer elastomers, and fluoroelastomer.

Response: Thanks to the reviewers for their comments. We performed the modification of polyurethane, waterborne polyurethane, styrene-ethylene-butylene-styrene, cyclic olefin copolymer elastomers, fluorine rubber and fluorine gum, and all the modifications have been successful except for cyclic olefin copolymer (COC). This may be due to its higher T_g (140-170°C), which makes it in a glassy state at room temperature. The specific data and pictures are shown below.

The AC and swelling ratio of different elastomers modified with PAA

Comparison of different elastomers before (upper) and after (lower) modification.

Comment No.6: The paper claims biocompatibility of the CEHH. However, no results have tested for biocompatibility. Please report cell viability results.

Response: Polyacrylic acid and silicone rubber are widely used in the biological field, and their

cytotoxicity studies have been reported, and have been proved to be unharmed. (Science Advances, 2021, 7, eabe8739, <https://doi.org/10.1126/sciadv.abe8739>; Biomaterials. 1990, 11(6), 393-6. [https://doi.org/10.1016/0142-9612\(90\)90093-6](https://doi.org/10.1016/0142-9612(90)90093-6).) The composite of silicone rubber and poly(acrylic acid) has also been used in biological fields such as cochlear implants due to its good biocompatibility. (Biomaterials, 1994, 15(14), 1161-1169, [https://doi.org/10.1016/0142-9612\(94\)90237-2](https://doi.org/10.1016/0142-9612(94)90237-2).) In general, both the single application of acrylic acid or silicone rubber, or the combination of the two, show good biocompatibility. Furthermore, we performed cytotoxicity experiments on calcified CEBH (MTT assay). As shown in the following figure, no significant difference was found when the leaching solution was added, proving that CEBH has good biocompatibility.

The viability of L929 cells co-cultured with the calcified CEBH for 24 h.

Comment No.7: The shape adaptability of CEHH for potential biomedical applications is demonstrated with ionic stimuli such as Ca^{2+} or citric acid. However, the concentrations utilized in the experiments (0.1M CaCl_2 acts as a body tissue irritant and is slightly toxic during ingestion https://www.mccsd.net/cms/lib/NY02208580/Centricity/Shared/Material_Safety_Data_Sheets_MSDS/_MSDS_Sheets_Calcium_Chloride_solution_0_1M_148_00.pdf) are much higher than physiologically relevant concentrations (~2mM calcium). Please demonstrate the shape adaptability at physiologically relevant concentrations.

Response: Thanks to the reviewers for their comments. We agree with this point. The higher concentration of Ca^{2+} may be toxic in vivo, but the Ca^{2+} solution is used to trigger the shape adaptability rather than maintain it. Therefore, the leakage of Ca^{2+} and the cell viability of calcinated hydrogel are what should be paid attention to. First, we record the Ca^{2+} leakage of the calcified CEBH sheet ($10 \times 10 \times 1 \text{ mm}^3$) immersed in 50 mL ultrapure water (shaken at a speed of 110/min), and the ion chromatography results showed that the concentration of calcium ion in ultrapure water increased by only 27 $\mu\text{g/mL}$. Furthermore, we performed cytotoxicity experiments on calcified CEBH (MTT assay, see SI for specific experimental steps) and no significant difference was found when the leaching solution was added, proving that CEBH has good biocompatibility.

Calcium ion released by calcified CEBH in 50 mL ultrapure water.

The viability of L929 cells co-cultured with the calcified CEBH for 24 h.

Comment No.8: The name elastomer/hydrogel hybrids is very confusing as it leads the reader to think of conventional elastomer/hydrogel bilayer hybrids. Renaming the material to reflect that the elastomer backbone itself becomes a hydrogel would provide clarification.

Response: Thanks to the reviewers for their comments. We rename the material as Connective-tissue inspired elastomer-based hydrogels (CEBH).

Reviewers' Comments:

Reviewer #1:

Remarks to the Author:

Thank you to the authors for their response.

Although the response was comprehensive, the manuscript still lacks in providing clear explanation for certain observations.

1. Figure 2a: a) Why does the grafting front increases only after AC > 30%?
b) From Figure S3b the swelling ratio increases with increasing AC. However, the same is not reflected in images shown in Figure 2a and could be misleading. For example, the sample with 0 wt.% AC after swelling in toluene, is smaller than the one with 13 wt.%, whereas.
c) "...the possibility of a bi-continuous phase structure can be completely excluded." However, from Figure 2a, the samples with 30 wt.% and 43 wt.% AC bend upon swelling. This is commonly observed for bi-layer structure due to differences in swelling ratios. From these images, it seems like there is a "bi-continuous phase" in these gels.
2. Figure 2b: What was the method used for measuring shore hardness of silicone? Was it indentation?
3. Figure 3: a) Puncture resistance is usually measured in terms of force/unit area using a static pressure test. How do these gels compare against the other puncture resistant hydrogels reported so far?
b) Do these gels show hysteresis upon repeated tensile tests? How does it vary as a function of AC?
4. Page 14: "...interaction between -COO- and Ca²⁺, and can recover its original shape in a dilute HCl". What kind of interaction helps in shape recovery?
5. Page 16: In general, wound dressings also lead to wound healing. Did the authors perform any in-vivo experiments proving the applicability of this hydrogel for wound dressing?

Reviewer #2:

Remarks to the Author:

The author has made detailed revisions and additions based on our comments. I have no concerns about the manuscript and recommend publication.

Reviewer #3:

Remarks to the Author:

The authors addressed all the issues the reviewers raised, so this reviewer recommend Nature Communications to publish the article.

Response to Reviewers' Comments

Response to the comments of Reviewer #1

Comment No.1: Figure 2a: a) Why does the grafting front increases only after AC > 30%?

b) From Figure S3b the swelling ratio increases with increasing AC. However, the same is not reflected in images shown in Figure 2a and could be misleading. For example, the sample with 0 wt.% AC after swelling in toluene, is smaller than the one with 13 wt.%, whereas.

c) "...the possibility of a bi-continuous phase structure can be completely excluded." However, from Figure 2a, the samples with 30 wt.% and 43 wt.% AC bend upon swelling. This is commonly observed for bi-layer structure due to differences in swelling ratios. From these images, it seems like there is a "bi-continuous phase" in these gels.

Response: Thanks for the reviewer's comments.

With the increment of AC, the grafting front gradually penetrates the elastomers till the silicone rubber is fully modified, and the expanding of different parts of CEBH in water (outside modified part) and toluene (inner unmodified layer) can illustrate this phenomenon. The authors apologize for the inconsistency in the previous version of figure. We repeated the experiments and strictly controlled the swelling process of the samples to ensure that they swelled under the same conditions. The new version of Figure 2a shows monotonic changes when the AC increases (Figure a).

The bending of the sample in Figure 2a is caused by the bending of silicone rubber during the modification process instead of anisotropic internal stress by bi-continuous phases, and using a larger container in the modification process can solve this problem. Besides, in our following work, we have printed a 3D-shape Chinese-dragon-shaped model with high precision using a DLP 3D printer, and after being modified into CEBH using the same technique, the details of the model maintained without bending or deformation, which can prove the penetrating grafting can reach a uniform state after being fully grafted (Figure b as follows).

(a) Digital images of CEBH with different acrylic acid contents after being swollen in 1 g/mL Cu^{2+} solution (left) and toluene (right). (b) Digital images of dragon-shaped CEBH in different states.

Comment No.2: Figure 2b: What was the method used for measuring shore hardness of silicone? Was it indentation?

Response: Thanks for the reviewer's comments. The Shore hardness data of the silicone rubbers in Figure 2b was given by the provider and verified by a hardness tester (hemuele, LX-A). The hardness tester uses the indentation method. The verification data was as follows.

The Shore's hardness of silicone rubbers verified by a hardness tester (hemuele, LX-A)

Comment No.3: Figure 3: a) Puncture resistance is usually measured in terms of force/unit area using a static pressure test. How do these gels compare against the other puncture resistant hydrogels reported so far?

b) Do these gels show hysteresis upon repeated tensile tests? How does it vary as a

function of AC?

Response: Thanks for the reviewer's comments.

a) The diameter of the needle in our puncture experiment is 1 mm, which conforms to the national standard of China (GB/T 37841-2019). To compare the anti-puncture ability of other hydrogels, we performed a puncture experiment with a diameter of 10 mm needle and observed obvious umbrella puncture deformation. The experimental data and test videos have been added to SI.

In the puncture experiment, the comparison before (left) and after (right) modification ((a) needle diameter 1mm, (b) needle diameter 10 mm).

However, making a comparison to puncture-resistant hydrogels reported in the literature is difficult. The first reason is that only a few literature use (quasi-)static puncture tests, and several researchers use sharp objects such as a screwdriver, (Sci. China Technol. Sci. 2021, 64, 827; Macromol. Rapid Commun. 2020, 2000185; Colloids and Surfaces A 2020, 589, 124402) nail/needle, (Adv. Funct. Mater. 2023, 33, 2304415;) or scissors (Ind. Eng. Chem. Res., 2023, 62, 18484.) to puncture on the hydrogel films to form an umbrella shape, which suggests that the films have good anti-puncture properties without any data. Compared with these, the umbrella shape of CEBH is sharper than most of them and should be considered as anti-puncture hydrogel. There are also quite a few researches that use static puncture tests. However various needles are used. We obtained as many reports as we could and tried to make a comparison with them. As shown in the following table, the CEBH in our work is much better than the ordinary PAAm hydrogel,⁴ and is 8 folds of the SA-AAm double network hydrogel,³ comparable with the montmorillonite reinforced hydrogel,⁶ and 1/8 folds of hydrogel composites laminated with aramid fabric.⁵ Objectively speaking, the puncture resistance performance of CEBH reported by this work is at the forefront of hydrogels except for fabric-reinforced hydrogels (whose anti-puncture properties were provided mainly by the fabrics).

Hydrogel	Load (N)	Needle diameter (mm)
BRC ¹	1.06*	0.3 (tip of sharp needle)
alginate hydrogels ²	1.2	\
SA-AAm ³	12	10
PAAm ⁴	0.5	1
PVA/SA/Gly hydrogel ^{5**}	57	1
s-BNCH ^{6***}	50	3 (tip of needle)
This work	96 (max)	10
	7.2 (max)	1

1. Chemical Engineering Journal **2023**, 454, 140261; 2. Progress in Biomaterials **2017**, 6 (4), 157; 3. Polymer Testing **2022**, 116, 107782; 4. Soft Matter **2015**, 11 (23), 4723. 5. Journal of Materials Research and Technology, 2022, 21, 2915; 6. Chemistry of Materials **2023**, 35 (15), 5809.

* Calculated from the needle diameter and pressure reported (15 MPa); ** Hydrogel composites laminated with aramid fabric; *** Montmorillonite reinforced hydrogel.

b) We performed repeated tensile tests of the CEBHs. When the CEBH is not totally modified, there is slightly hysteresis, but no hysteresis is observed in the fully modified ones. The hysteresis of CEBH decreases with the increase of AC.

Repeated tensile curves of different AC of CEBH after complete swelling (36 %, 48 %, 56 % from left to right).

Comment No.4: Page 14: "...interaction between -COO^- and Ca^{2+} , and can recover its original shape in a dilute HCl". What kind of interaction helps in shape recovery?

Response: Thanks for the reviewer's comments. The H^+ ions can exchange with the Ca^{2+} and cause the decomposition of a complex of -COO^- and Ca^{2+} . Therefore, the shape fixing caused by the crosslinking points of Ca^{2+} can be released and help the hydrogel recover its permanent shape. This strategy is widely applied in polymers with carboxyl such as alginate, EDTA, pH, and higher concentrations of monovalent ions (such as Na^+), etc. can also release the crosslinking agent Ca^{2+} . (International Journal of Polymeric Materials and Polymeric Biomaterials, 69(4), 230-247)

Comment No.5: Page 16: In general, wound dressings also lead to wound healing. Did the authors perform any in-vivo experiments proving the applicability of this hydrogel for wound dressing?

Response: Thanks for the reviewer's comments. We performed wound healing experiments with rats, and the results are as follows. The wound healing rates of the control group (treated with Tegaderm™ film) and model group (treated with CEBH) were similar, and there are no significant differences in the wound closure results, showing that the CEBH has similar effects to Tegaderm™ films on the wound healing of experiment rats. The Masson and HE staining images of wound sections on day 22 from both control and model groups are also shown in Figure d, showing that the CEBH hydrogel didn't introduce additional skin fibrosis or inflammatory cells. Therefore, the CEBH promoted wound healing similarly to Tegaderm™ films and showed good biosafety properties.

(a) Representative photographs of skin wounds treated with Tegaderm film (control), CEBH on days 0, 4, 8, 12, 16, 20 and 22. (b, c) Traces of wound closure on days 0, 2, 4, 6, 8, 10, 12, 14, 16, 18, 20 and 22. (d) Masson's and HE staining of wound sections obtained from control and model groups on day 22.

Reviewers' Comments:

Reviewer #1:

Remarks to the Author:

Thank you to the authors for addressing all of our concerns.